# Low-Dose Sorafenib Promotes Cancer Stem Cell Expansion and Accelerated Tumor Progression in Soft Tissue Sarcomas

**DOI:** 10.3390/ijms25063351

**Published:** 2024-03-15

**Authors:** Sylvia M. Cruz, Khurshid R. Iranpur, Sean J. Judge, Erik Ames, Ian R. Sturgill, Lauren E. Farley, Morgan A. Darrow, Jiwon Sarah Crowley, Arta M. Monjazeb, William J. Murphy, Robert J. Canter

**Affiliations:** 1Division of Surgical Oncology, Department of Surgery, University of California Davis, Sacramento, CA 95817, USA; 2Department of Pathology, Stanford University, Stanford, CA 94305, USA; 3Bioinformatics and Computational Biology, University of North Carolina at Chapel Hill, Chapel Hill, NC 27599, USA; 4Department of Pathology and Laboratory Medicine, University of California Davis, Sacramento, CA 95817, USA; 5Department of Radiation Oncology, University of California Davis, Sacramento, CA 95817, USA; 6Department of Dermatology, University of California Davis, Sacramento, CA 95817, USA; wmjmurphy@ucdavis.edu

**Keywords:** cancer stem cells, sorafenib, ALDH, sarcoma, survival

## Abstract

The cancer stem cell (CSC) hypothesis postulates that heterogeneous human cancers harbor a population of stem-like cells which are resistant to cytotoxic therapies, thus providing a reservoir of relapse following conventional therapies like chemotherapy and radiation (RT). CSCs have been observed in multiple human cancers, and their presence has been correlated with worse clinical outcomes. Here, we sought to evaluate the impact of drug dosing of the multi-tyrosine kinase inhibitor, sorafenib, on CSC and non-CSCs in soft tissue sarcoma (STS) models, hypothesizing differential effects of sorafenib based on dose and target cell population. In vitro, human cancer cell lines and primary STS from surgical specimens were exposed to escalating doses of sorafenib to determine cell viability and expression of CSC marker aldehyde dehydrogenase (ALDH). In vivo, ALDH^bright^ CSCs were isolated, exposed to sorafenib, and xenograft growth and survival analyses were performed. We observed that sarcoma CSCs appear to paradoxically respond to the tyrosine kinase inhibitor sorafenib at low doses with increased proliferation and stem-like function of CSCs, whereas anti-viability effects dominated at higher doses. Importantly, STS patients receiving neoadjuvant sorafenib and RT on a clinical trial (NCT00864032) showed increased CSCs post therapy, and higher ALDH scores post therapy were associated with worse metastasis-free survival. These data suggest that low-dose sorafenib may promote the CSC phenotype in STS with clinically significant effects, including increased tumor growth and higher rates of metastasis formation in sarcoma patients.

## 1. Introduction

Cancer stem cells (CSCs) have been identified as a clinically relevant subpopulation within multiple cancers in both pre-clinical models and patient samples [1,2,3]. CSCs are characterized as quiescent cells within heterogeneous tumors with their dormancy promoting resistance to standard cytotoxic treatments such as chemotherapy and radiotherapy (RT) [1,4,5,6]. The presence of CSCs has been associated with worse clinical outcomes across multiple cancer types [6,7,8,9]. Although the etiology of CSCs remains controversial and debate continues regarding the origin of CSCs, both genetic and epigenetic mechanisms have been implicated in the phenotype and function of stem-like tumor cells. These factors make them distinct from bulk tumor cells and clinically relevant as a sub-population [6,8,10]. Moreover, CSCs are an important target in cancer therapy due to their ability to repopulate and promote relapse in breast, sarcoma, pancreas, renal, and other cancers [11,12,13,14]. These stem-like properties, such as self-renewal, differentiation potential, and upregulation of key signaling pathways (such as Notch, Hedgehog, Hippo-YAP) have been linked with repopulation of tumors post treatment [15,16,17,18]. This suggests that targeting CSCs could significantly improve clinical outcomes for cancer patients.

We previously demonstrated that tyrosine kinase inhibitors (TKIs), especially sorafenib, are associated with CSC enrichment because of their depleting effects on bulk tumor cells [19,20,21]. Sorafenib is a TKI that has shown promise as an anti-neoplastic inhibitor of B-Raf and Raf1/c-Raf signaling pathways [22]. Sorafenib is FDA-approved for renal cell and hepatocellular carcinoma based on improvements in both progression-free and overall survival [23,24]. We previously evaluated sorafenib with RT in a phase I trial of neoadjuvant therapy for soft tissue sarcoma (STS), establishing a maximum tolerated dose [25]. However, given our pre-clinical data showing differential effects of sorafenib on CSC and non-CSC populations, we sought to determine the potential pre-clinical and clinical impact of these CSC changes mediated by sorafenib on outcomes. We used pazopanib, a clinically approved STS treatment, as a relevant control [26]. We hypothesized that CSCs respond differently to pharmacologic stresses given their different genetic and epigenetic compositions. We sought to evaluate the impact of sorafenib on CSC proliferation, CSC function, and clinical outcomes using multiple in vitro and in vivo models including primary STS specimens from surgery and patients receiving neoadjuvant sorafenib and RT on a clinical trial (NCT00864032).

## 2. Results

### 2.1. Sorafenib Stimulates CSC Expansion In Vitro at Low Doses in Sarcoma Cell Lines with Anti-Viability Effects at Higher Doses

Although the exact origin of CSCs remains controversial, the CSC phenotype has been reproducibly identified by multiple experimental techniques [27,28,29]. In STS, high expression of the enzyme aldehyde dehydrogenase (so-called ALDH^bright^) has been linked with the CSC phenotype while other markers (CD24, CD44, and CD133) have shown mixed results [30,31,32,33]. Given the plasticity associated with CSC markers [34,35], we sought to revalidate the functional properties of ALDH^bright^ sarcoma CSCs with the prototypical CSC colony outgrowth assay. Using sorted ALDH^bright^ cells from Ewing’s sarcoma cell line A673, we assessed colony forming units (CFUs) and observed significantly higher numbers of colonies among the ALDH^bright^ versus ALDH^dim^ populations in a dose-dependent fashion (Appendix A).

We then assessed the effect of sorafenib on CSC and non-CSC populations in vitro. A673, SK-LMS, and SW982 cell lines were exposed to increasing doses of sorafenib. We observed a consistent cytotoxic effect of sorafenib at sorafenib doses of 8 μM and higher (Figure 1A–C). However, there was a significant increase in CSCs at doses ranging from 2 to 4 μM as observed in the absolute number of ALDH^bright^ cells (Figure 1D–F). At doses above 8 μM, this increase in ALDH^bright^ absolute cell numbers was reversed, and the number of ALDH^bright^ cells declined significantly (*p* < 0.001). Additionally, we observed no differences in ALDH^bright^ cell numbers over the range of doses for pazopanib (Figure 1D–F), consistent with the primarily anti-angiogenic, non-cytotoxic effects of this agent [26]. Representative flow cytometry is shown in Figure 1G,H. Importantly, as shown in Figure 1I,J, ALDH^bright^ cells increased in both numbers and percentages at a dose of 4 μM sorafenib, whereas ALDH^bright^ cell numbers decreased at high-dose sorafenib (32 μM), but the percentage of ALDH^bright^ cells increased, consistent with differential effects of the drug on bright and dim populations at the different doses.

Since drug dosing in patients frequently involves long-term, continuous exposure, we tested the effect of longer-term exposure to sorafenib on the viability and frequency of both CSC and non-CSC populations in vitro. As shown in Appendix A, A673 cells were continuously cultured in the presence of 4 μM sorafenib which was refreshed every 3–4 days for 14 days. After this 14-day period, cells were trypsinized, washed, and replated for 48 h sorafenib exposure in vitro. Cells grown in standard culture conditions (without chronic sorafenib exposure) reproduced a similar pattern of an approximate doubling of numbers of ALDH^bright^ cells at low doses from 2– to 4 μM, which was statistically significant (*p* ≤ 0.001, Appendix A). Similarly, ALDH^bright^ A673 cells cultured under standard conditions and then acutely exposed to sorafenib again significantly declined in numbers of ALDH^bright^ cells at doses ≥8 μM (*p* < 0.001). Although the cells chronically exposed to sorafenib had a less exaggerated increase in ALDH^bright^ numbers after acute sorafenib exposure, this increase in ALDH^bright^ cell numbers was statistically significant at low doses while anti-viability effects were observed in both CSCs and non-CSCs at doses ≥8 μM (Appendix A). Taken together, these data suggest that low-dose sorafenib leads to an increase of ALDH^bright^ CSCs in sarcoma lines, which is modulated at higher doses due to cytotoxic effects of sorafenib.

### 2.2. Low-Dose Sorafenib Stimulates In Vitro Proliferation of CSCs across Multiple Non-Sarcoma Tumor Cell Lines

To determine whether these sorafenib effects were specific to STS, we assessed other tumor lines. Similar to STS, we detected comparable effects of sorafenib in vitro on human pancreas, breast, and renal cell lines with CSC-promoting effects at low doses but anti-viability effects at doses ≥16 μM (Figure 2A–F). Additionally, we assessed the effects of sorafenib using other CSC markers since over-reliance on individual CSC markers, like ALDH, has been criticized [33]. For PANC-1, we assessed CD24, CD44, and EpCam since these have been linked with a more specific phenotype for CSCs in pancreas cancer (Figure 2G–I) [36,37]. We saw a similar pattern of an increase in the absolute number of CD24^+^CD44^+^EpCAM^+^ CSCs at low-dose sorafenib (*p* ≤ 0.01). As before, at doses ≥8 μM, we observed significant decreases in CSCs, again suggesting dose-dependent effects of sorafenib with CSC expansion at low doses and anti-viability effects at higher doses, further emphasizing that low-dose sorafenib effects on CSC enrichment extend beyond sarcomas and is not restricted to ALDH^bright^ as the CSC marker.

### 2.3. Sorafenib Stimulates Proliferation of Human Primary Sarcoma CSCs Ex Vivo

To better represent the tumor heterogeneity of the in vivo microenvironment [38], we evaluated the effects of sorafenib on primary STS tumors. Tumor cells from surgical specimen were processed into single-cell suspension [19,30,39] for ex vivo treatment with increasing doses of sorafenib. Representative H&E photomicrographs are shown (Figure 3A,C). Following ex vivo sorafenib exposure for 24 h, a dedifferentiated liposarcoma specimen showed stable numbers of ALDH^bright^ cells at doses ranging from 0 to 8 µM, a statistically significant increase in ALDH^bright^ cells at 16 μM, and a decrease to baseline at 32 μM (Figure 3B). A leiomyosarcoma primary tumor also showed a significant increase in ALDH^bright^ cells at doses of sorafenib ≥16 μM (Figure 3D). In a third STS patient, we observed an increase in ALDH^bright^ cells from 4.1% to 14.5% after 24 h exposure to 4 μM of sorafenib ex vivo (Figure 3E) with representative flow cytometry of negative diethylaminobenzaldehyde (DEAB) control and sorafenib populations shown (Figure 3F). Although there was greater variability in the results with primary STS tumors following sorafenib exposure, we nevertheless observed similar effects with increased number and frequency of ALDH^bright^ CSCs in a dose-dependent fashion. Together, these data suggest that sorafenib is also associated with CSC-promoting effects at lower doses in primary STS tumors.

### 2.4. Low-Dose Sorafenib Promotes the CSC Phenotype In Vitro and In Vivo

We next sought to determine the impact of low-dose sorafenib exposure on the stem-like behavior of sarcoma CSCs in vitro by colony forming unit (CFU) evaluation. Exposure of A673, SK-LMS, and SW982 sarcoma lines to 4 µM sorafenib significantly increased CFU formation in all three cell lines (*p* ≤ 0.01), while pazopanib pre-treatment had no effect (Figure 4A–C). We then assessed the effect of sorafenib pre-treatment on in vivo xenograft formation. As shown in Figure 4D, 48 h pre-treatment of A673 cells with 4 μM sorafenib prior to implantation in NSG mice led to more rapid tumor growth in vivo in sorafenib-treated cells with approximately 10-day shorter median survival (*p* = 0.05, Figure 4E). Overall, these data indicate that exposure of sarcoma cells to low-dose sorafenib in vitro increases their stem-like behavior as evidenced by increased CFUs in vitro and xenograft growth in vivo, both of which are associated with the CSC phenotype.

### 2.5. ALDH^bright^ A673 Cells Expand after Low-Dose Sorafenib while ALDH^dim^ Cells Do Not

To better assess the specific effects on CSCs versus non-CSCs, we evaluated the effects of sorafenib treatment on sorted ALDH^bright^ and ALDH^dim^ A673 populations. Sorted cells were cultured in sorafenib 4 µM or vehicle (Figure 5A) for 48 h. As shown in Figure 5B,C, we observed that the ALDH^bright^ cells increased significantly in both frequency and numbers when exposed to low-dose sorafenib compared to ALDH^dim^ cells. Sorted ALDH^bright^ cells gave rise to both ALDH^bright^ and ALDH^dim^ cells, while sorted ALDH^dim^ cells gave rise to essentially all ALDH^dim^ cells. We also observed that the anti-viability effects of low-dose sorafenib (4 µM) significantly impacted the frequency and absolute numbers of ALDH^dim^ cells while there was no significant anti-viability effect of 4 µM sorafenib on ALDH^bright^ cell frequencies or absolute numbers (Figure 5D,E). In a separate experiment, as shown in the schema in Figure 5F, we assessed in vivo xenograft formation using sorted A673 ALDH^bright^ and ALDH^dim^ cells that were not exposed to sorafenib. The sorted A673 ALDH^bright^ and ALDH^dim^ cells were immediately injected into the opposite hind limbs of NSG mice, confirming the differential ability for in vivo tumor formation between the CSC and non-CSC populations (Figure 5G,H). 

### 2.6. Association of ALDH Enrichment with Worse Oncologic Outcomes in Sorafenib/RT-Treated Patients Treated on a Clinical Trial

Given the potential clinical implications of these findings, we evaluated archived STS patient specimens to link our in vitro and in vivo results with clinical samples. We previously showed that neoadjuvant sorafenib in combination with RT is tolerable with some evidence of activity in locally advanced extremity STS (NCT#00805727) [25]. Using immunohistochemistry (IHC) on archived specimens from this clinical trial (N = 8) and historical control patients treated with neoadjuvant RT alone (N = 13), we sought to evaluate ALDH scores pre- and post neoadjuvant therapy and assess whether ALDH scores were associated with clinical outcomes. As shown in Figure 6A, there was a significant increase in ALDH scores following neoadjuvant therapy among sorafenib patients who developed metastases. In contrast, there was no statistical difference in ALDH scores among RT monotherapy patients who developed metastases versus those who did not (Figure 6B). We observed a strong inverse correlation between increasing fold change in ALDH score and worse MFS (r = −0.746, *p* = 0.03, Figure 6C) among sorafenib/RT patients, whereas there was no significant association between increasing fold change in ALDH score and MFS (r = −0.193, *p* = 0.53) among STS patients treated with neoadjuvant RT monotherapy (Figure 6D). We further stratified patients by fold change in ALDH score after neoadjuvant therapy, using the median value of five as the cut point, and compared MFS among patients in the respective cohorts (Figure 6E–G). MFS was worse among sorafenib/RT patients with ALDH fold change ≥5 compared to those with ALDH fold change <5. Given the small cohort, this difference approached but did not reach statistical significance (*p* = 0.09). In contrast, MFS was not different in patients treated with RT alone when stratified by changes in ALDH scores (*p* = 0.33). These data suggest that increased ALDH scores are associated with worse oncological outcomes in STS patients treated with sorafenib/RT whereas a similar pattern was not observed in patients treated with neoadjuvant RT alone.

Lastly, we sought to investigate potential mechanistic pathways underlying the association of low-dose sorafenib with CSC expansion and worse clinical outcomes. We again sorted A673 cells to obtain purified ALDH^bright^ populations and exposed them to low-dose sorafenib (4 μM) or vehicle for 48 h. Using multiplex qPCR for the expression of cell cycle and proliferation genes, we observed a statistically significant upregulation of proliferation markers *PCNA* and *EGFR* in the sorafenib-treated ALDH^bright^ cells (Figure 6H). Given this significant increase in *EGFR*, we evaluated serum growth factors by Luminex Assay from STS patients treated with surgery only, neoadjuvant RT only, and combination RT/sorafenib. Although not statistically significant, we observed an increased fold change in serum epidermal growth factor (EGF) in patients who were treated with sorafenib and RT compared to the other cohorts (Figure 6I), potentially implicating the EGF-EGFR pathway [40].

## 3. Discussion

Our data show a consistent increase in cells with a CSC phenotype when exposed to low doses of the multi-tyrosine kinase inhibitor sorafenib. These observations were made in sarcoma cell lines, as well as pancreas, breast, and renal cancer cell lines. These results were recapitulated in primary human sarcoma samples and archived specimens from a phase I clinical trial combining sorafenib with RT as neoadjuvant therapy. These data suggest that low-dose sorafenib promotes expansion of a CSC population that appears to have implications for clinical outcomes.

CSCs have been implicated in multiple cancer processes, including repopulation following cytotoxic therapies, epithelial-to-mesenchymal transition, and tumor recurrence after dormancy [41,42]. In aggregate, the evidence suggests that CSCs are a clinically relevant subpopulation of malignant cells in the bulk tumor. These quiescent cells are resistant to standard cancer treatments such as chemotherapy and RT [1,2,3]. Consequently, targeting CSCs is an unmet need in cancer therapy. From our results, the increase in ALDH^bright^ cells across models implies that low-dose sorafenib has the potential to have the opposite intended effect and supports the hypothesis that low-dose sorafenib increases the frequency of CSCs in heterogeneous tumor populations, especially STS.

A strength of our study was the access to STS samples and archived clinical trial samples, both of which are important in CSC studies since the nature of cell culture in adherent conditions can limit translational relevance. Additionally, we observed that sorted ALDH^bright^ A673 cells expanded and had increased viability when treated with low-dose sorafenib, while ALDH^dim^ A673 cells did not. The increase in ALDH^bright^ cells across models implies that low-dose sorafenib has the potential to have unintended effects besides anti-viability/anti-proliferative effects. These results also support the theory that low-dose sorafenib may increase the frequency of CSCs in a heterogenous tumor population.

Regarding the mechanism of sorafenib enhancement, we noted that sorafenib-treated A673 cells had an increase in proliferative markers *Ki67, PCNA*, and *EGFR* by PCR. We also observed that mice injected with sorafenib-treated cells, compared to cells grown in standard culture conditions, had a significantly shorter survival time in vivo, secondary to tumor progression. This suggests that low-dose sorafenib increases activation and proliferation in CSCs with greater progression of bulk tumor. These data are supported by a recent study investigating BRaf and MEK inhibitors in hepatocellular carcinoma where the authors observed that administration of a low concentration of sorafenib leads to sustained, and in some cases increased, MAPK signaling via B- and CRaf dimerization [43]. MAPK signaling has often been shown to promote tumor progression [44]. While we did not look at MAPK in our analysis, we observed that there was an increase in proliferative markers *EGFR* and *PCNA* by PCR and EGF in serum in patients who received sorafenib/RT, which may enhance the proliferation of tumor cells [45].

Furthermore, the potential for striking differential effects of TKIs on discrete cancer cell populations was highlighted in a case report of a patient with melanoma administered the selective RAF inhibitor vemurafenib. After 11 days of treatment, a previously palpable subcutaneous melanoma tumor regressed, but the patient soon developed progression of a previously occult chronic myelomonocytic leukemia [46]. The acute development and rapid progression of leukemia resolved after cessation of TKI therapy. This study provides a fascinating proof-of-concept demonstration that pharmacological agents can have differential effects on disparate signaling pathways depending on both the cell type and the genetic and epigenetic machinery which can vary in different cellular constituents within the same patient.

Despite our provocative data, it is important to acknowledge the limitations of our study. For example, we lack in-depth data addressing mechanistic insights that cause low-dose sorafenib to increase the number of ALDH^bright^ cells and promote the CSC phenotype. Although our data implicate low-dose sorafenib in increased numbers of CSCs in vitro, which appears to be related to CSC proliferation rather than conversion to a stem-like phenotype, the precise molecular pathways remain unresolved. Similarly, although we observed striking data that patients with greater increases in ALDH expression after treatment with sorafenib on a clinical trial had worse clinical outcomes, pharmacokinetic data are not available for the on-trial patients, and thus it is not possible to show that the higher ALDH scores and worse oncological outcomes are related to low-dose sorafenib in the serum. Since drug absorption and metabolism may be affected by weight, biological sex, age, and other variables, it is possible that patients may be exposed to drug levels in vivo which can have heterogeneous or disparate effects. However, the potential for drug levels to vary across time and tissues in patients further reinforces the relevance of our findings since fluctuations in drug levels may lead to unintended consequences, as our data highlight. Taken together, our data reinforce the potential for differential and sometimes paradoxical effects of cancer therapeutics based on dose and the possibility for these effects to be different among CSC and non-CSC cancer cell populations.

## 4. Materials and Methods

### 4.1. Cell Lines

Cancer cell lines (A673 Ewing’s sarcoma, SW-982 synovial sarcoma, SK-LMS leiomyosarcoma, PANC-1 pancreas, MDA-MB231 breast, and ACHN renal) were obtained from the American Type Culture Collection (Manassas, VA, USA) and maintained in recommended tissue culture media as described previously [19,30,39]. Cell line authentication was verified every 6–12 months using short tandem repeat analysis.

### 4.2. Colony Forming Unit Assay

Soft agar was purchased from BD Biosciences (San Jose, CA, USA) and plated on tissue culture plates from Corning Inc. (Corning, NY, USA) as described previously [47].

### 4.3. Flow Cytometry and Sorting

ALDEFLUOR™ expression (STEMCELL Technologies, Vancouver, BC, Canada) was detected using commercial kits as described previously [19,30,39] and included FITC fluorescent channel for the detection of ALDH. PE-Cy7 anti-human CD24, Pacific Blue anti-human CD44, and APC-Cy7 anti-human EpCAM antibodies were purchased from BioLegend (San Diego, CA, USA). 7-Aminoactinomycin D (7-AAD) viability dye was used to identify dead cells. All samples were acquired on an LSR Fortessa with HTS (BD Biosciences, San Jose, CA, USA) and analyzed with FlowJo software, version 10.8.1 (Tree Star, Ashland, OR, USA).

### 4.4. Primary Tumor Samples

Primary STS tumor tissue (CCS15-10, CCS15-12, and SA-0982) was obtained fresh from surgical specimens through the UC Davis Cancer Center Biorepository. Informed consent was obtained from all patients on an Institutional Review Board-approved protocol. Tumor samples were processed into single-cell suspensions for CSC phenotyping and ex vivo exposure to TKIs as described previously [19,30,39,48,49].

### 4.5. Retrospective Analysis of Clinical Trial

Results from a phase I trial of neoadjuvant radiotherapy plus sorafenib for patients with locally advanced STS of the extremity and body wall were reported previously (Clinical Trial Information: NCT#00805727) [25]. Here, clinical outcomes were reanalyzed with respect to ALDH staining on archived tumor samples, with further details in Appendix A.

### 4.6. In Vivo Experiments

Female NOD.Cg-Prkdcscid Il2rgtm1Wjl/SzJ (NSG) mice aged 6–8 weeks were obtained from the Jackson Laboratory (Bar Harbor, ME, USA). Mice were housed under specific pathogen-free conditions in AAALAC-approved vivarium. Briefly, 1 × 10^4^ sorted A673 ALDH^bright^ and ALDH^dim^ cells were immediately injected subcutaneously into the contralateral hind limbs of NSG mice (within 60 min of cell sorting) and tumor growth was measured. In separate experiments, A673 cells were treated with 4 μM sorafenib or vehicle for 48 h and then injected subcutaneously into the flanks of NSG mice. Animals were monitored for tumor growth and survival using humane endpoints. All experimental protocols were approved by the UC Davis Institutional Animal Care and Use Committee. The study is reported in accordance with ARRIVE guidelines.

### 4.7. H&E Staining and Immunohistochemistry

Archived tumor samples were stained for ALDH1 (BD Transduction Laboratories, San Jose, CA, USA) as described previously [39,50]. H-scores were determined in a blinded fashion (MAD).

### 4.8. Multiplex qRT-PCR

RNA was extracted from A673 ALDH^bright^ cells that were treated with 4 µM sorafenib using RNeasy Mini kits (Qiagen, Germantown, MD, USA,) according to manufacturer’s protocol. The RNA was then reverse transcribed to cDNA using the High-Capacity cDNA Reverse Transcription Kit from Applied Biosystems (Vilnius, Lithuania). Gene-specific primers were designed using Primer-BLAST. Quantitative real time-PCR was performed using the RT2 SYBR Green Mastermix (Qiagen, Germantown, MD, USA) by way of the StepOnePlusTM Real-Time PCR system (Applied Biosystems, Vilnius, Lithuania). RNA amplification and analysis were performed as described previously [39].

### 4.9. Luminex Assay

Multiplex analysis using R&D Luminex technology was used to determine plasma levels of growth and angiogenic factors before and after treatment for patients as indicated. Samples were measured in duplicates. Additional details regarding reagents and resources used can be found in Appendix A. 

### 4.10. Statistics

Summary statistics were reported as mean ± standard error or median (range) where appropriate. Categorical variables were compared using a chi-squared test. Parametric and non-parametric statistics were used as indicated. For comparison of more than 2 groups, statistical significance was determined using a one-way ANOVA followed by a Bonferroni multiple-group comparison test. Survival curves were evaluated using the Kaplan–Meier method. ALDH scores before and after treatment were analyzed using the two-sided paired *t*-test. Statistical analyses were performed using Graph-Pad Prism 8. Significance was set at *p* < 0.05.

## Figures and Tables

**Figure 1 ijms-25-03351-f001:**
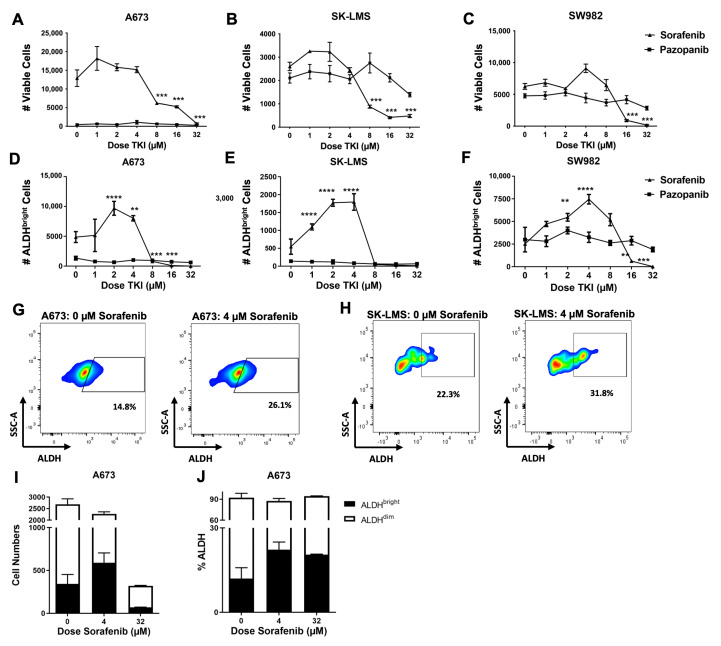
Sorafenib treatment is associated with increased cancer stem cells at low doses, but cytotoxic effects at higher doses in sarcoma cells in vitro. (**A**–**C**) A673 Ewing’s sarcoma cells, SK-LMS leiomyosarcoma cells, and SW982 synovial sarcoma cells were cultured with increasing doses of sorafenib and pazopanib. Absolute numbers of viable cells were measured by flow cytometry using 7-AAD to distinguish viable and non-viable cells. (**D**–**F**) A673 Ewing’s sarcoma cells, SK-LMS leiomyosarcoma cells, and SW982 synovial sarcoma cells were cultured with increasing doses of sorafenib and pazopanib, and ALDH^bright^ cells were measured by flow cytometry using the ALDEFLUOR™ assay. (**G**) Representative flow cytometry staining of A673 ALDH^bright^ cells comparing no treatment (left) to 4 µM sorafenib treatment (right). (**H**) Representative flow cytometry staining of SK-LMS ALDH^bright^ cells comparing no treatment (left) to 4 µM sorafenib treatment (right). (**I**) Cell numbers of ALDH^bright^ and ALDH^dim^ cells within untreated (0 µM), low-dose sorafenib (4 µM), and high-dose sorafenib (16 µM) groups were measured by flow cytometry using the ALDEFLUOR™ assay. (**J**) Percentage of ALDH^bright^ and ALDH^dim^ cells within untreated (0 µM), low-dose sorafenib (4 µM), and high-dose sorafenib (16 µM) groups were measured by flow cytometry using the ALDEFLUOR™ assay. All experiments were performed in triplicate. ** *p* < 0.01, *** *p* < 0.001, **** *p* ≤ 0.0001 via one-way ANOVA with Tukey’s test.

**Figure 2 ijms-25-03351-f002:**
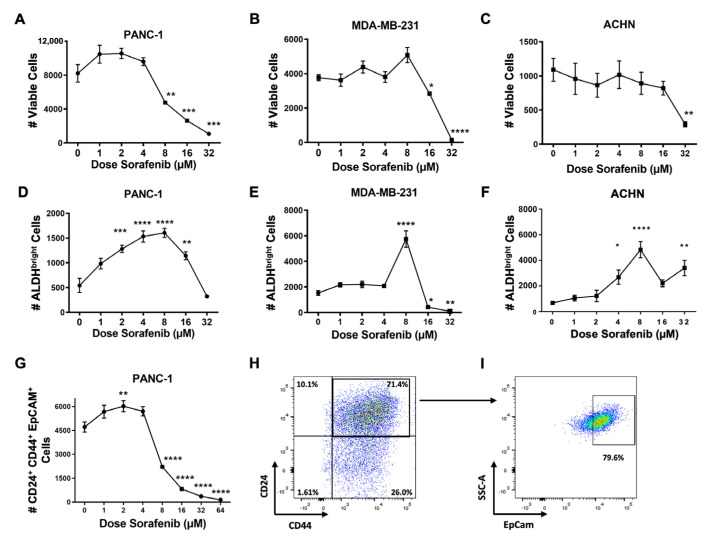
Effects of sorafenib in vitro on non-sarcoma cancer lines. (**A**–**C**) Human pancreas cancer (PANC-1) cells, breast cancer (MDA-MB-231) cells, and renal cell cancer (ACHN) cells were cultured with increasing doses of sorafenib. Absolute numbers of viable cells were measured by flow cytometry using 7-AAD to distinguish viable and non-viable cells. (**D**–**F**) Human pancreas cancer PANC-1 cells, breast cancer MDA-MB-231 cells, and renal cell cancer ACHN cells were cultured with increasing doses of sorafenib, and ALDH^bright^ cells were measured by flow cytometry using the ALDEFLUOR™ assay. (**G**) Human pancreas cancer PANC-1 cells were cultured with increasing doses of sorafenib, and CD24, CD44, and EpCam surface marker expression was measured by flow cytometry. (**H**,**I**) Representative flow cytometry staining of PANC-1 cells showing CD24 versus CD44 parent gating with EpCam^+^ cells from the CD24^+^CD44^+^ double positive gate at 2 µM of sorafenib. All experiments were performed in triplicate. * *p* < 0.05, ** *p* < 0.01, *** *p* < 0.001, **** *p* ≤ 0.0001 via one-way ANOVA with Tukey’s post hoc test.

**Figure 3 ijms-25-03351-f003:**
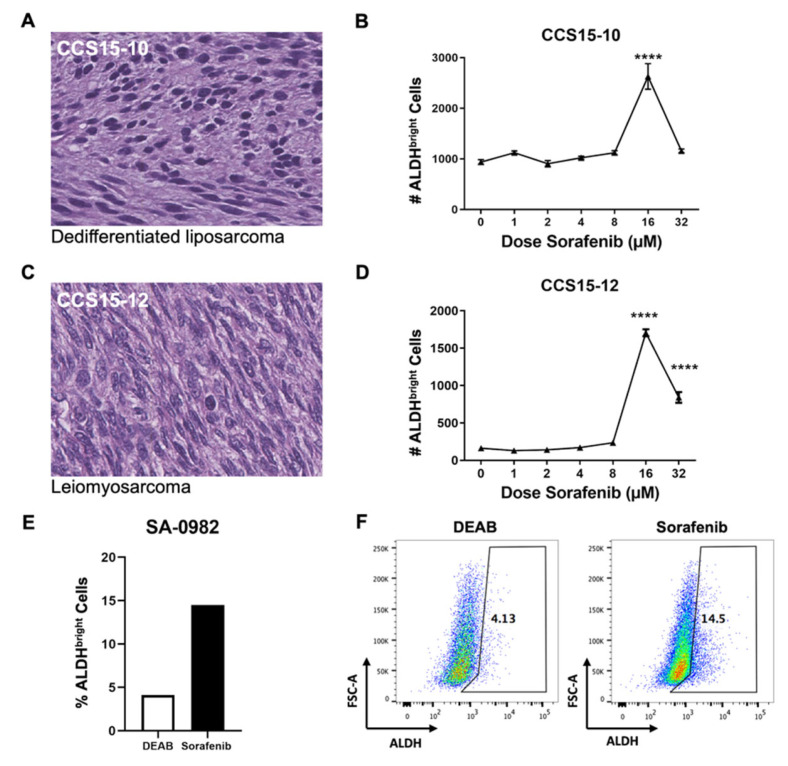
Effects of sorafenib ex vivo on heterogeneous primary sarcoma specimens. (**A**) Representative H&E stain from human sarcoma patient CCS15-10 showing malignant mesenchymal cells consistent with dedifferentiated liposarcoma at approximately 300× magnification. (**B**) Fresh surgically resected tissue was processed into single-cell suspension and plated ex vivo with indicated doses of sorafenib, then analyzed by flow cytometry for ALDH expression using the ALDEFLUOR™ assay. (**C**) Representative H&E stain from human sarcoma patient CCS15-12 showing malignant spindle cells consistent with leiomyosarcoma at approximately 300× magnification. (**D**) Fresh surgically resected tissue was processed into single-cell suspension and plated ex vivo with indicated doses of sorafenib, then analyzed by flow cytometry for ALDH expression using the ALDEFLUOR™ assay. (**E**) Sarcoma patient SA-0982 underwent surgical resection, and fresh tumor tissue was processed into a single-cell suspension and plated overnight with sorafenib at 4 µM compared to diethylaminobenzaldehyde (DEAB) control. Samples were then analyzed for ALDH expression by flow cytometry using the ALDEFLUOR™ assay. (**F**) Representative flow cytometry staining of ALDH^bright^ cells comparing DEAB control (left) to sorafenib-treated cells (right). **** *p* ≤ 0.0001 via one-way ANOVA with Tukey’s post hoc test compared to dose level 0.

**Figure 4 ijms-25-03351-f004:**
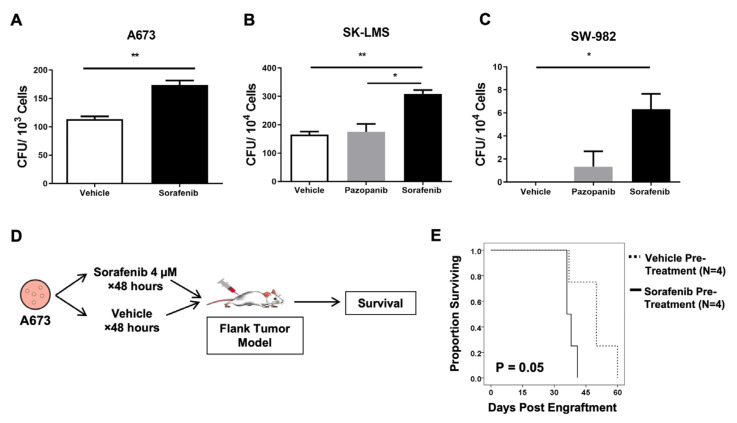
Functional effects of low-dose sorafenib. (**A**–**C**) A673 Ewing’s sarcoma cells, SK-LMS leiomyosarcoma cells, and SW982 synovial sarcoma cells were cultured overnight with 4 µM of sorafenib, pazopanib, or vehicle and then plated in soft agar plates for growth in non-adherent culture conditions. Colony forming units were counted and scored. (**D**) Schema shows experimental design for in vivo assessment of tumor growth following sorafenib incubation in vitro. Sarcoma cells were incubated with and without low-dose sorafenib for 48 h, subcutaneously injected into the flanks of NSG mice and allowed to form tumors. Tumor growth was measured, and survival was determined according to humane endpoints. (**E**) Mice pre-treated with low-dose sorafenib prior to tumor implantation had a significantly shorter survival time compared to placebo-treated mice (*p* = 0.05, by log-rank test). An in vivo experiment was performed twice. In vitro experiments were performed in triplicate. * *p* < 0.05, ** *p* < 0.01 via one-way ANOVA with Tukey’s post hoc test.

**Figure 5 ijms-25-03351-f005:**
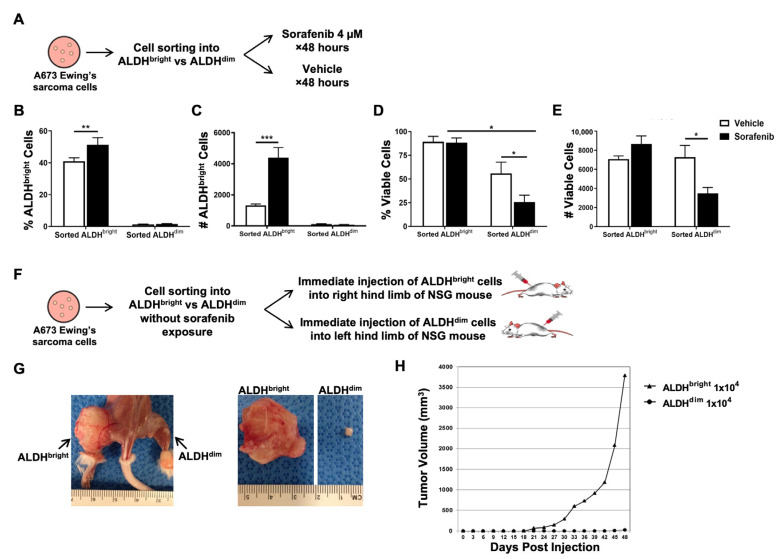
Effects of sorafenib on sorted ALDH^bright^ and ALDH^dim^ subpopulations in A673 Ewing’s sarcoma cells. (**A**) Schema depicting a 48 h in vitro experiment. A673 Ewing’s sarcoma cells were sorted via fluorescence-activated cell sorting (FACS) into ALDH^bright^ and ALDH^dim^ subpopulations and plated with and without low-dose sorafenib. The number of ALDH^bright^ cells and viable cells were then measured by flow cytometry. (**B**,**C**) Percentage and cell numbers of the ALDH^bright^ cells within treated and untreated sorted ALDH^bright^ and ALDH^dim^ subpopulations were measured by flow cytometry using the ALDEFLUOR™ assay. (**D**,**E**) Percentage and cell numbers of viable cells within treated and untreated sorted ALDH^bright^ and ALDH^dim^ subpopulations were measured by flow cytometry using 7-AAD. (**F**) Schema depicting an in vivo experiment. A673 Ewing’s sarcoma cells were sorted via FACS into ALDH^bright^ and ALDH^dim^ subpopulations without sorafenib exposure and immediately injected subcutaneously into hind limbs of NSG mice. (**G**) Representative images of xenograft formation after injection of sorted ALDH^bright^ and ALDH^dim^ cells into hind limbs of NSG mice. (**H**) Mice treated with ALDH^bright^ A673 cells had a larger tumor volume than those treated with ALDH^dim^. All experiments were performed in triplicate. * *p* < 0.05, ** *p* < 0.01, *** *p* < 0.001 via one-way ANOVA with Tukey’s post hoc test.

**Figure 6 ijms-25-03351-f006:**
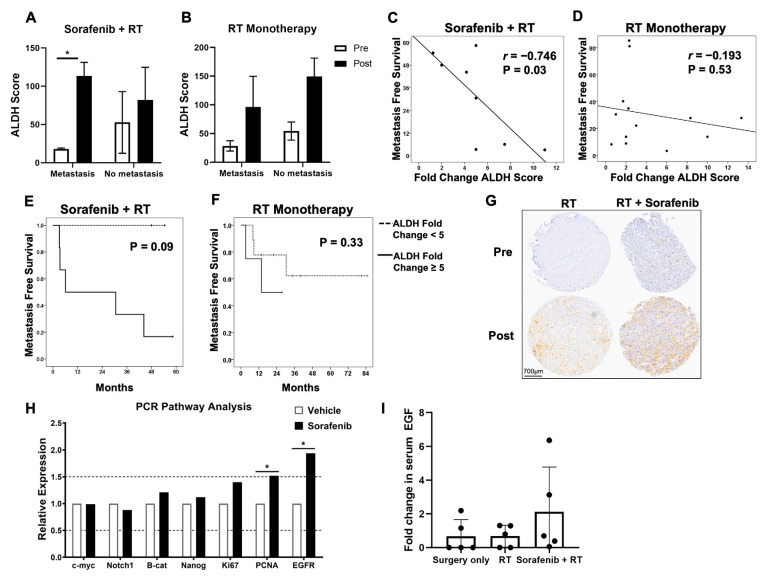
Outcomes of sorafenib and RT-treated patients. (**A**,**B**) Sarcoma patients received treatment of either sorafenib + RT (N = 8) or RT monotherapy (N = 13), and ALDH score was assessed in both groups pre- and post therapy by immunohistochemistry. Patients were stratified by whether they developed metastasis or not, and sorafenib + RT patients who developed metastases had a significant increase in ALDH scores whereas other groups did not. (**C**,**D**) Pre- and post-therapy ALDH scores were used to calculate fold change in ALDH score, and fold change ALDH score was then correlated to metastasis-free survival in sorafenib + RT versus RT monotherapy cohorts. There was a strong and statistically significant negative correlation between increasing ALDH score and metastasis-free survival in sorafenib + RT patients, while there was a weak and insignificant negative correlation between increasing ALDH score and metastasis-free survival in RT monotherapy patients. (**E**,**F**) Kaplan–Meier survival analysis shows metastasis-free survival stratified by ALDH fold change score (using median fold change of 5 as the stratification factor) in sorafenib + RT and RT monotherapy cohorts. (**G**) Representative immunohistochemical staining for ALDH on sarcoma patient tumor tissue at diagnosis (pre) and at surgical resection post neoadjuvant therapy (RT monotherapy versus RT + sorafenib). (**H**) FACS-sorted A673 ALDH^bright^ cells were exposed to 4 µM sorafenib or vehicle for 48 h, and multiplex qPCR was performed for expression of cell cycle and proliferative gene transcripts. *PCNA* and *EGFR* expression was significantly increased in sorafenib-treated cells compared to vehicle. (**I**) Fold change in serum levels of EGF from diagnosis to surgery in sarcoma patients treated with either surgery only, RT monotherapy, or sorafenib + RT. PCR and Luminex assays were performed in duplicates. * *p* < 0.05 via one-way ANOVA with Tukey’s post hoc test.

## Data Availability

The datasets generated during and/or analyzed during the current study are available from the corresponding author on reasonable request.

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
