# Peer review of "Low-Dose Sorafenib Promotes Cancer Stem Cell Expansion and Accelerated Tumor Progression in Soft Tissue Sarcomas"

_ijms, 2024, doi:10.3390/ijms25063351_

Round 1

Reviewer 1 Report

Comments and Suggestions for Authors

Cruz et.al. discovered that low doses of the tyrosine kinase inhibitor sorafenib promote increased proliferation of cancer stem cells and tumor progression in soft tissue sarcomas. They performed in vitro experiments using various types of human cancer cell lines and in vivo studies, demonstrating xenograft growth and survival analyses. Additionally, they observed an increase in ALDH after administering a low dose of sorafenib. The authors also illustrated the association of ALDH with a worse prognosis in neoadjuvant sorafenib treatment for patients treated in a clinical trial.

The findings presented in this work are both interesting and important in understanding the outcomes of different doses of sorafenib in patients. The manuscript is well-written and described. I recommend the publication of this manuscript in its current form.

Author Response

Cruz et.al. discovered that low doses of the tyrosine kinase inhibitor sorafenib promote increased proliferation of cancer stem cells and tumor progression in soft tissue sarcomas. They performed in vitro experiments using various types of human cancer cell lines and in vivo studies, demonstrating xenograft growth and survival analyses. Additionally, they observed an increase in ALDH after administering a low dose of sorafenib. The authors also illustrated the association of ALDH with a worse prognosis in neoadjuvant sorafenib treatment for patients treated in a clinical trial.

The findings presented in this work are both interesting and important in understanding the outcomes of different doses of sorafenib in patients. The manuscript is well-written and described. I recommend the publication of this manuscript in its current form.

We thank the reviewer for taking the time to review our manuscript. We appreciate their overall favorable assessment of our manuscript.

Reviewer 2 Report

Comments and Suggestions for Authors

The authors Cruz et al. have produced a thoroughly detailed, elegant, and clearly written study, describing the effect of low-dose sorafenib on cancer stem cell expansion. The methodology is meticulously described, and the text is thoughtfully written, allowing for a clear understanding of the study's procedures and results. Additionally, the reproducibility of the authors' findings in in vitro and ex vivo systems, as well as archival samples provide robust evidence for their findings. These findings highlight significant implications of sorafenib use in STS management.

In reviewing the manuscript, several points for clarification and refinement have been identified:

  1. 1. Regarding the results, it would be beneficial to include dose-response curves for cell lines treated with pazopanib. This would allow for unambiguous interpretations of the results showing low and unchanged numbers of ALDHbright cells in the pazopanib-treated cells, relative to the total cell population under this treatment group.

  2. 2. The authors' assertion regarding the primarily "anti-angiogenic, non-cytotoxic effects of this agent" in Figure 1D-F could be strengthened with additional data on total cytotoxicity for pazopanib-treated cells, as mentioned in the previous point.

  3. 3. Considering the significant difference in cell viability between sorted

  4. ALDHbright and ALDHdim cells, was this difference corrected for when injecting these cells into the NSG mice for the ex vivo assays? If so, please strengthen your arguments by stating as such. If not, please clarify and include this caveat in the text.

  5. 4. There seems to be a discrepancy between the number of cases presented in Figure 6D (13 dots or cases) and the N=11 described in the text. Please clarify/check the figure.

  6. 5. Please include details of the multiplex qPCR in the methodology section.

Overall, I would like to once again commend the authors on a well-written and elegant series of experiments. 

Comments on the Quality of English Language

The manuscript is clearly written. Nevertheless, kindly check on the following:

1.       Ensure terms like in vitro, ex vivo, and in vivo are italicized consistently throughout the text.

2.       Punctuation edit “but The” on line 100.  

Author Response

The authors Cruz et al. have produced a thoroughly detailed, elegant, and clearly written study, describing the effect of low-dose sorafenib on cancer stem cell expansion. The methodology is meticulously described, and the text is thoughtfully written, allowing for a clear understanding of the study's procedures and results. Additionally, the reproducibility of the authors' findings in in vitro and ex vivo systems, as well as archival samples provide robust evidence for their findings. These findings highlight significant implications of sorafenib use in STS management.

In reviewing the manuscript, several points for clarification and refinement have been identified:

  1. Regarding the results, it would be beneficial to include dose-response curves for cell lines treated with pazopanib. This would allow for unambiguous interpretations of the results showing low and unchanged numbers of ALDHbright cells in the pazopanib-treated cells, relative to the total cell population under this treatment group.

The reviewer makes a cogent point that the dose response of pazopanib would provide important information. We apologize for not including in the original manuscript and have revised Figures 1A-C to show pazopanib alongside sorafenib curves. We believe this provides important context regarding the differences for clinical uses of the TKIs.

  1. The authors' assertion regarding the primarily "anti-angiogenic, non-cytotoxic effects of this agent" in Figure 1D-F could be strengthened with additional data on total cytotoxicity for pazopanib-treated cells, as mentioned in the previous point.

As above discussed above, the reviewer raises a good point regarding the importance of the pazopanib data to allow for unambiguous interpretation of our results. Although the mechanisms of action of the TKIs are viewed as being different, there can be significant overlap of how these agents works. As noted above, we have provided data regarding the effect of pazopanib on viable cell counts in Figures 1A-C. We show cumulative cytotoxicity based on changes in the numbers of total live cells in a dose response curve, though we recognize that this data could alternatively be shown by the number of dead cells on a dose response curve. We hope this addresses the reviewer’s concern.

  1. Considering the significant difference in cell viability between sorted ALDHbright and ALDHdim cells, was this difference corrected for when injecting these cells into the NSG mice for the ex vivo assays? If so, please strengthen your arguments by stating as such. If not, please clarify and include this caveat in the text.

The reviewer makes an excellent point and is correct that viability of the ALDHdim cells does not appear to sustain and is decreased after 48 hours of treatment of the sorted cells, as compared to the ALDHbright cells. Given that we injected the cells into the NSG mice within 60 minutes of cell sorting, we did not reassess viability by hemocytometry count but recognize that this could be a confounding factor. We appreciate the reviewer pointing this out and have added clarifying text to the manuscript.

  1. There seems to be a discrepancy between the number of cases presented in Figure 6D (13 dots or cases) and the N=11 described in the text. Please clarify/check the figure.

We appreciate the reviewer’s attention to detail and apologize for this oversight. We have adjusted the text to correspond appropriately with the data presented.

  1. Please include details of the multiplex qPCR in the methodology section.

We have added further details regarding the multiplex qPCR in the methods section.

Overall, I would like to once again commend the authors on a well-written and elegant series of experiments.

We thank the reviewer for taking the time to review our manuscript as well as for their overall positive assessment. We appreciate the opportunity to address the reviewer’s concerns above.

Comments on the Quality of English Language

The manuscript is clearly written. Nevertheless, kindly check on the following:

  1. Ensure terms like in vitro, ex vivo, and in vivo are italicized consistently throughout the text.
  2. Punctuation edit “but The” on line 100.

We appreciate these comments and have edited the manuscript as suggested.